# The level of health anxiety before and during the COVID-19 pandemic

**Anja Davis Norbye**[1]*, **Erlend Hoftun Farbu**[1], **Christoffer Lilja Terjesen**[1,2], **Nils Fleten**[3], **Anje Christina Höper**[1,4]

**1** Department of Community Medicine, UiT the Arctic University of Norway, Tromsø, Norway, **2** Department of Rehabilitation, University Hospital of North Norway, Tromsø, Norway, **3** Norwegian Labour and Welfare Administration Troms and Finnmark, Tromsø, Norway, **4** Department of Occupational and Environmental Medicine, University Hospital of North Norway, Tromsø, Norway

* anja.davis.norbye@uit.no

## Abstract

### Background

Concerns about disease and an increase in health anxiety levels are expected consequences of the COVID-19 pandemic. However, there have been few longitudinal studies of health anxiety in the general population during this time period. The aim of this study was to examine health anxiety levels before and during the COVID-19 pandemic in an adult, working population in Norway.

### Material and methods

This study included 1012 participants aged 18–70 years with one or more measurements of health anxiety (1402 measurements total) from the pre-pandemic period (2015 to March 11, 2020) and/or during the COVID-19 pandemic (March 12, 2020 to March 31, 2022). Health anxiety was measured with the revised version of the Whiteley Index-6 scale (WI-6-R). We estimated the effect of the COVID-19 pandemic on health anxiety scores with a general estimation equation analysis, and age, gender, education, and friendship were included in subgroup analyses.

### Results

We found no significant change in health anxiety scores during the COVID-19 pandemic compared to the pre-pandemic period in our adult, working population. A sensitivity analysis restricted to participants with two or more measurements showed similar results. Moreover, the effect of the COVID-19 pandemic on health anxiety scores was not significant in any subgroup analysis.

### Conclusion

Health anxiety remained stable, with no significant change observed between the pre-pandemic period and the first 2 years of the COVID-19 pandemic in an adult, working population in Norway.

**Data Availability Statement:** This study uses data from two larger surveys, the HIW study and the Tromsø study. According to Norwegian law, both are considered sensitive data including health information. De-identified data sets are not

considered anonymous data sets in Norway due to the large number of variables to each ID that can make it possible to backtrace despite de-identification. Therefore, there are legal restrictions to publicly share the dataset in relation to publication. However, the data set from the Tromsø study can be applied for access for researchers by contacting the project committee of the Tromsø study: tromsous@uit.no.

**Funding:** Helse Nord Health Trust funded the costs related to linkage of data between information of those participating both in the Tromsø study and Health in Work study. The publication charges for this article were funded by a grant from the publication fund of UiT The Arctic University of Norway. The main project Health in Work was funded by: - The research and development fund of the Norwegian Labour and Welfare Administration (2018 announcement) - Contributions from the Helse Nord Health Trust - Contributions from UiT The Arctic University of Norway. The funders had no role in study design, data collection and analysis, decision to publish, or preparation of the manuscript.

**Competing interests:** The authors have declared that no competing interests exists.

## Introduction

Health anxiety (HA) can be defined as a worry of having or getting a disease, ranging on a continuum from mild worry to excessive anxiety [1,2]. High HA has been associated with higher healthcare use [3] and risk of sick leave [4], and it is also associated with a decreased health-related quality of life [5]. During the COVID-19 pandemic, the population has been instructed to be aware of respiratory symptoms, to get tested if symptoms develop, and to self-isolate in case of symptoms or a positive test. As HA is characterized by bodily preoccupation and reassurance-seeking behaviour [6], researchers have speculated that the pandemic both increased the risk of developing HA [7], and triggered underlying HA [8]. Although several measurement tools for HA exist, the most commonly used to screen for HA in the general population is the Whiteley Index [9].

Several studies have reported on the negative impacts of the COVID-19 pandemic on mental health, including increased prevalence of both depression and anxiety [10–12]. Younger age groups and women in particular were found to have worse mental health during the pandemic [10–12]. However, a longitudinal study from Germany found that symptoms of anxiety, depression, and loneliness were highest in the first few months of the pandemic, before they stabilized and declined [13], a possible indication of an adaptation to the situation.

Many researchers expected to see elevated or high HA levels in the general population during the pandemic, and especially in proposed vulnerable populations such as students and people with chronic diseases [14,15]. This has indeed been observed in several cross-sectional studies [14–19]. However, few longitudinal studies have examined trends in HA levels during this period. To our knowledge, only one study from Denmark reported longitudinal data with a baseline before the start of the pandemic [20]. They found only minor changes in median HA scores before and during the first wave of the pandemic. We lack knowledge on longitudinal trends in HA, especially after the initial phases of the COVID-19 pandemic. Therefore, the aim of this study was to examine HA levels in the periods before and during the COVID-19 pandemic in an adult, working population in Norway, thereby improving the scarce level of knowledge on the topic.

## Material and methods

### Data sources

This study was conducted with a longitudinal design, using data from the Health in Work (HIW) study. The HIW study is an ongoing, pragmatic, cluster-randomized controlled trial examining the effect of different workplace interventions. It includes a working population (18–70 years of age, working 20% or more) in Troms and Finnmark, the northernmost county in Norway. Details of the HIW study are described elsewhere [21]. Briefly, participants were recruited from June 2019 through June 2021; all participants completed two identical questionnaires that collected information on a wide range of health- and work-related variables. The first questionnaire was completed at recruitment (Q1) and the second 12 months thereafter (Q2). For participants living in the municipality of Tromsø, HIW data was linked to data from the Tromsø study, a population-based survey which was first conducted in the 1970s [22]. The present analysis included data from the 7th and most recent wave of the Tromsø study (Tromsø7), that was conducted in 2015–2016.

### Variables

**Outcome variable.** The Tromsø7 questionnaire, HIW Q1, and HIW Q2 all included a revised version of the Whiteley Index-6 (WI-6-R), which has shown satisfactory psychometric

**Table 1. Questions included in the revised Whiteley Index-6.**

| Item number | Text |
| --- | --- |
| 1 | Do you think there is something seriously wrong with your body? |
| 2 | Do you worry a lot about your health? |
| 3 | Is it hard for you to believe the doctor when he/she tells you there is nothing to worry about? |
| 4 | Do you often worry about the possibility that you have a serious illness? |
| 5 | If a disease is brought to your attention (e.g., via TV, radio, internet, newspapers, or someone you know), do you worry about getting it yourself? |
| 6 | Do you have recurring thoughts about being ill that are difficult to get off your mind? |

Each question was answered using a 5-point Likert Scale (0 = "not at all", 1 = "to some extent", 2 = "moderately", 3 = "to a considerable extent", or 4 = "to a great extent"). Responses to all questions were then summed for a total HA score, which ranged from 0–24, where 24 points indicated the highest possible HA level.

properties [23] and which was used in the longitudinal study by Petersen et al [20]. Questions included in the WI-6-R are outlined in Table 1.

**Covariables.** Information on covariables was taken from the Tromsø7, and the HIW study questionnaires. As this study was based on data from the HIW study, which included an intervention that aimed to reduce HA, we adjusted for the intervention in the models.

Increasing age has been considered a risk factor for adverse outcomes of COVID-19. No clear gender differences have been found in HA levels [24] or in the prevalence of high HA [25,26] in the general population before 2020, but some studies have found gender differences in HA levels during the pandemic [16,27]. Therefore, we included age and gender as covariables.

Socioeconomic status may have affected HA levels during the COVID-19 pandemic. As we did not have data regarding participants' income, we used educational level as a proxy for socioeconomic status. Educational level was assessed by the question: "What is the highest level of education you have completed?", with the response options: primary education up to 10 years of schooling, vocational/upper secondary education, or university/college education.

We also included a variable concerning friendship, as friendship has been associated with HA level [24], and loneliness has been associated with higher HA level during the COVID-19 pandemic [28,29]. Friendship was assessed by two questions: "Do you have enough friends who can give you help and support when you need it?" and "Do you have enough friends you can talk confidentially with?". Friendship was coded as "no", for those who answered "no" to both questions; "to some extent", for those who answered "yes" to only one question; and "yes", for those who answered "yes" to both questions.

Finally, the WI-6-R in HIW Q1 and HIW Q2 contained the introduction "have you in the last 12 months. . ." for each question, in line with the validated questionnaire. Unfortunately, this introduction was omitted in the WI-6-R contained in the Tromsø7 questionnaire. As we lack this timeframe for HA levels in the Tromsø7 data, we included a dichotomous variable indicating that the HA score came from Tromsø 7 in our models.

## Statistical analysis

The World Health Organization declared a worldwide pandemic on March 11, 2020, and Norway declared lockdown on March 12, 2020. Therefore, the pre-pandemic period was defined as 2015 to March 11, 2020, and the period during the COVID-19 pandemic was defined as March 12, 2020 to the end of the study period (March 31, 2022). All observations with missing

data on the WI-6-R, age or gender were excluded. We also excluded observations if there was uncertainty whether Q1 was completed before COVID-19.

As higher levels of lockdown have been associated with worse mental health in the United Kingdom [30], we divided the study period into six smaller time periods in the descriptive analysis, chosen based on changes in restrictions throughout the pandemic [31]. Time period 1 was 2015 to 2016, corresponding to pre-pandemic measurements from Tromsø7. Time period 2 was June 2019 to March 11, 2020, corresponding to pre-pandemic measurements for participants who answered HIW Q1. Time period 3 was March 12, 2020 to August 31, 2020, when Norway was on lockdown with relatively comprehensive restrictions. Time period 4 was September 1, 2020 to October 31, 2020, when most restrictions in Norway were lifted. Time period 5 was November 1, 2020 to June 30, 2021, when Norway re-introduced restrictions as the second wave of the pandemic started. Time period 6 was July 1, 2021 to March 31, 2022, when fewer restrictions were in place.

To calculate the effect of the COVID-19 pandemic, we used the questionnaires (Tromsø7, HIW Q1, and HIW Q2) as time points, and created a dichotomous variable indicating whether the questionnaire was completed in the pre-pandemic period or during the COVID-19 pandemic. We used general estimation equation (GEE) analyses [32] with a gamma distribution and log-link function to model the population average effect of the pandemic, as this is a recommended approach to estimate the population average effect [33]. We used an exchangeable correlation structure and included all participants with at least one measurement of HA in the GEE analyses and adjusted for the intervention, age, gender, and whether the score came from Tromsø7. We then estimated the change in HA score between the pre-pandemic period and during the COVID-19 pandemic.

As a sensitivity analysis, we repeated the GEE model, but restricted it to participants with two or more measurements of HA. We also ran analyses including interaction-terms between COVID-19 and age, gender, educational level and friendship to see if COVID-19 had a greater impact in subgroups.

All statistical analyses were performed in Stata 17.

## Ethics

This study was approved by the Regional Committee for Medical and Health Research Ethics (ID 2018/2262). All participants gave written informed consent before inclusion, and gave explicit consent for linkage to the Tromsø study data when relevant.

## Results

### Study sample and participant characteristics

The participating workplaces were within the following categories: healthcare (23), the educational system (25), service- and industry (17), and office workplaces (31). Of the 1383 persons participating in the HIW study, 1012 were included in the present analyses. By the end of this study period, 915 participants answered HIW Q1, 360 had answered HIW Q2, and linkage to Tromsø7 data was possible for 127. Therefore, the final study sample consisted of 1402 measurements of HA. The mean age of participants as of January 2019 was 42.4 (standard deviation: 11.6) and 75% were women. Most had university or college education (Table 2).

The mean(median) HA score was 2.19(1) out of 24 points. There were no considerable changes in HA levels during the six time periods investigated in the descriptive analysis (Fig 1).

In the GEE model for effect, we allowed for different combinations (Table 3).

**Table 2. Demographic and social characteristics of included study participants from the Health in Work (HIW) study.**

| Variable | Categories | N | Percent |
|---|---|---|---|
| Intervention at Q2 | HIW intervention | 147 | 59 |
| | Control group | 213 | 41 |
| | Missing | 0 | |
| Gender | Female | 757 | 75 |
| | Male | 255 | 25 |
| | Missing | 0 | |
| Educational level | Primary (≤10 years) | 48 | 5 |
| | Vocational/upper secondary | 313 | 31 |
| | University/college | 645 | 64 |
| | Missing | 6 | <1 |
| Friendship at Q1 | No | 78 | 8 |
| | To some extent | 80 | 8 |
| | Yes | 750 | 74 |
| | Missing | 104 | 10 |

## Estimated change in health anxiety between the pre-pandemic period and during the COVID-19 pandemic

In our study sample, we found no significant change in HA score between the pre-pandemic period and during the COVID-19 pandemic (Table 4). As a sensitivity analysis, we repeated

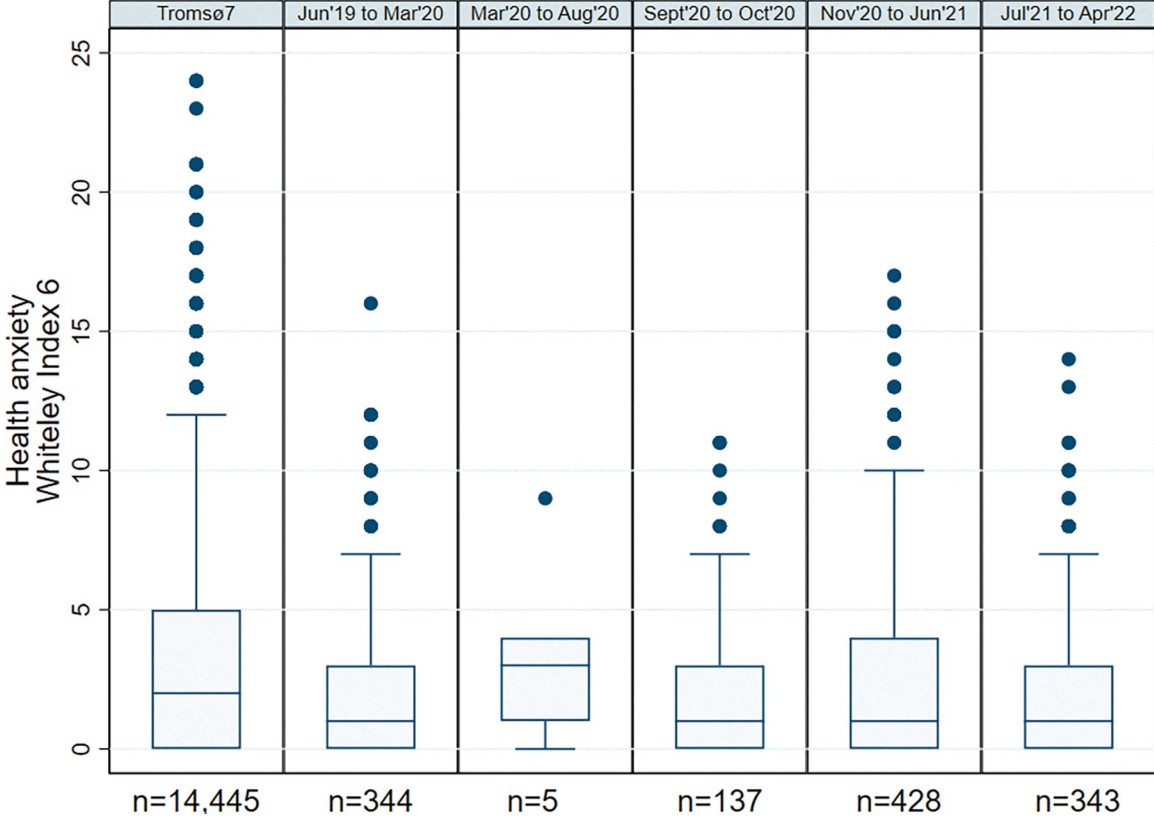

**Fig 1. Boxplots showing the distribution of WI-6-R scores during six different time periods.** 9 measurements from Q2 did not have an exact timepoint, and are therefore excluded from this figure (n = 1 393).

**Table 3. Overview of the different combinations and number of repeated measurements of health anxiety (HA).**

| | Tromsø7 | Q1 before March 12, 2020 | Q1 after March 12, 2020 | Q2 | N |
|---|---|---|---|---|---|
| N participants with one measurement | | x | | | 204 |
| | | | X | | 385 |
| | | | | x | 76 |
| | | | | | 665 |
| N participants having answered before and after March 12th, 2020 | X | x | | x | 29 |
| | X | | X | x | 10 |
| | | x | | x | 97 |
| | X | | X | | 54 |
| | X | | | x | 20 |
| | | | | | 210 |
| N participants having two or more measurements before or after March 2020 | X | x | | | 14 |
| | | | X | x | 129 |
| | | | | | 143 |
| Total N of participants | | | | | 1018 |
| Total N of HA measurements of HA | | | | | 1402 |

Q1: Questionnaire 1 in the Health in Work study; Q2: Questionnaire 2 in the Health in Work study.

the GEE model from the main analysis, but restricted it to participants with two or more measurements of HA and found similar results (estimated change 0.06 points) (Table 4). To explore whether different subgroups had different HA scores in response to the COVID-19 pandemic, we included interaction terms between COVID-19 and age, gender, educational level, and friendship. The analyses showed no significant change in HA score in any subgroup.

## Discussion

The aim of this study was to examine the HA levels in an adult population in the periods before and during the COVID-19 pandemic. In contrast with many cross-sectional studies [14–19], but in accordance with the results from Petersen et al. [20], we found no significant change in HA score in HIW study participants between the pre-pandemic period and during the COVID-19 pandemic. This was consistent when restricting the analysis to participants with two or more measurements, and when looking at subgroups of age, gender, educational level, and social network factors.

**Table 4. The estimated effect of the COVID-19 pandemic on health anxiety (HA) scores from a general estimation equation model.**

| | Estimated change in WI-6-R score | 95% CI | | |
|---|---|---|---|---|
| | | Lower | Upper | |
| **Main analysis[1]** | | | | $n_{individuals} = 1012$ |
| Effect of the pandemic | 0.25 | -0.06 | 0.57 | $n_{measurements} = 1402$ |
| **Sensitivity analysis[2]** | | | | $n_{individuals} = 353$ |
| Effect of the pandemic | 0.06 | -0.34 | 0.46 | $n_{measurements} = 743$ |

[1]: Change in HA score between the pre-pandemic period and during the COVID-19 pandemic. Adjusted for the intervention, age, gender and whether the score came from Tromsø7.

[2]: Main analysis repeated among participants with two or more measurements of HA.

WI-6-R: Revised Whiteley Index-6; CI: Confidence interval.

Our results of no significant change in HA between the pre-pandemic period and during the COVID-19 pandemic contrasts those of studies that have reported an increase in HA. There are several possible reasons for this disparity. First, cross-sectional [14,15,17] and retrospective [16,18,19] studies are more prone to recall bias regarding health [34]. Secondly, most studies gathered their data during the initial period of the pandemic, in the spring of 2020, and most looked at mental health indicators in general. A meta-analysis [35] on mental health during the pandemic found that, although there was a slight increase in depression and anxiety immediately after the pandemic was declared, the levels gradually declined to pre-pandemic levels. One study from Germany examined the perceived change in mental health during the initial phase of the pandemic and found that, whereas most people reported an increase in mental health symptoms compared to before the pandemic, only a minority reported a change in HA [34]. Similarly, in Denmark, a repeated cross-sectional study found that, whereas COVID-related worries and loneliness fluctuated according to number of cases and governmental restrictions, measures of anxiety and mental health remained stable [36]. Taken together, there is increasing evidence that, on a population level, mental health in general and HA in particular, did not increase during the first 2 years of the COVID-19 pandemic. Indeed, some authors have proposed that HA is more of a stable trait that is not affected to any large degree by external events [37]. They proposed more studies that examine HA as a risk factor for unfavourable effects of the pandemic, not as an outcome in itself [37].

Although few studies have investigated longitudinal observations of HA before and during the COVID-19 pandemic, one can speculate that there are cultural and between-country differences that may impact our results. Some have argued that, in Norway, the COVID-19 pandemic has been perceived as more of a societal crisis than a health crisis; that people were more worried about the economic and other non-health-related consequences of the pandemic than about the disease itself [38]. In Norway, both the number of cases and the mortality rate were lower than in many European countries [39]. National aid packages gave economic support to those who had to stay at home due to symptoms or quarantine, and there were no reports of overwhelmed healthcare systems, in contrast to some other European countries. A longitudinal study reported that a high level of confidence in health personnel and satisfaction with health information were strong protective factors against poor mental health during the pandemic [40]. Several authors have found that concise information and effective COVID-19-related policy responses have had anxiety-reducing effects [41,42]. The presence of these factors in Norway might have contributed to the lack of significant change we observed in HA levels during the COVID-19 pandemic.

## Strengths and limitations

One strength of this study is the use of the WI-6-R as a validated measurement tool for HA, which makes comparison to results in pre-pandemic years possible. A questionnaire designed to capture specific concerns related to the pandemic could capture more specific worries, but it would have less comparability to questionnaires designed before the COVID-19 pandemic. However, we cannot conclude that the WI-6-R, which was originally designed to identify people with hypochondriasis and was later used mostly to identify those with severe HA, will enhance a construct like COVID-19-related anxiety in the same manner, although COVID-19-related anxiety has been proposed as a sub-group of the broader HA construct [8]. We did not include COVIC-19-specific questions, such as perceived risk of contracting the virus or of having COVID-19 in close social network and family. In the longitudinal study by Petersen et al [20], participants reported elevated worries in these aspects, but with limited increase in HA as a result. Therefore, we do not believe that including such variables would alter our results.

Another important strength of this study is that the HIW questionnaires collected information on a wide range of health- and work-related items, and did not have identification of HA as its primary aim when including participants. This can minimize the risk of selection bias compared to studies that recruited participants for the express purpose of examining HA. One study [43] examining risk factors for mental health during the pandemic reported high levels of anxiety and depression in their participants, which may indicate that people feeling an additional burden due to the pandemic are more likely to participate in COVID-19-related studies.

One limitation is the relatively few participants with two or more measurements (n = 353); more repeated measurements would have enhanced our confidence in the results. Some of the defined time periods in Fig 1 were wide, and it can be speculated that eventual spikes in HA would not be captured due to the time periods stated in this project. However, large discrepancies would have been seen as it would have affected the population mean. It is also important to note that this study was conducted in the working-age population, with working participants. Also, the HIW study and its intervention excluded participants who were employed less than 20% and who did not speak Norwegian. Although people on sick leave were not excluded from the study, it likely excluded people who may be most prone to worry and to socially isolate because of the pandemic (i.e. students, people on disability benefits, or those not in the work force for other reasons). Indeed, as the HIW study examines the work environment, participating workplaces might have been skewed to workplaces most interested in this topic. Some studies have found higher HA and other mental health strains in student populations [14] and populations with chronic diseases [15,44], and a national report [45] highlighted that some immigrant populations in Norway were overrepresented in the number of COVID-19 cases and in cases of serious disease, which could be partially explained by a lack of proficiency in the Norwegian language [45]. As a consequence, our study might have excluded some groups of the Norwegian population most affected by the pandemic–and who might have different HA levels.

Despite the present limitations, we found no significant change in HA scores associated with the COVID-19 pandemic in an adult, working population in Norway. In contrast to several cross-sectional reports [14–19] and expectations of increased HA [7,8] in the general population, our results are in accordance with both longitudinal studies on HA [20] and on mental health in general [35], indicating that HA remained stable during a period that, for many, has been unprecedented.

Our results can have implications to add to knowledge to public health decision-making and prioritising of healthcare support in times of pandemics. Perhaps other consequences of COVID-19, such as social and economic burdens, require targeted intervention over anxiety about health. In a larger context, COVID-19 has been one of the 21st century challenges, alongside the climate crisis and the Russian-Ukrainian war. We do not know the long-lasting impact of these challenges, but one study found that these concerns were correlated with higher anxiety and depression, especially among young adults [46]. As anxiety in general is moderately overlapping with HA [47], this could have an impact on the mean HA in the general population in years to come. Further research on health- and other long-term consequences in the general population is still warranted.

## Acknowledgments

The authors thank the study participants for providing data for the analyses.

## Author Contributions

**Conceptualization:** Anja Davis Norbye, Erlend Hoftun Farbu, Christoffer Lilja Terjesen, Nils Fleten, Anje Christina Höper.

**Data curation:** Erlend Hoftun Farbu, Christoffer Lilja Terjesen.

**Formal analysis:** Anja Davis Norbye, Erlend Hoftun Farbu, Christoffer Lilja Terjesen.

**Methodology:** Erlend Hoftun Farbu.

**Project administration:** Christoffer Lilja Terjesen, Nils Fleten, Anje Christina Höper.

**Writing – original draft:** Anja Davis Norbye.

**Writing – review & editing:** Erlend Hoftun Farbu, Christoffer Lilja Terjesen, Nils Fleten, Anje Christina Höper.

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
