## [Decision Letter · Decision Letter 0]

13 Mar 2023

PONE-D-23-01947The level of health anxiety before and during the COVID-19 pandemicPLOS ONE

Dear Dr. Norbye,

Thank you for submitting your manuscript to PLOS ONE. After careful consideration, we feel that it has merit but does not fully meet PLOS ONE’s publication criteria as it currently stands. Therefore, we invite you to submit a revised version of the manuscript that addresses the points raised during the review process.

We look forward to receiving your revised manuscript.

Kind regards,

Academic Editor

PLOS ONE

Journal Requirements:

Additional Editor Comments:

The article is interesting and may provide important information for public health interventions at local level, but I suggest to compare these results with other experiences that consider aside COVID 19, other cofactors associated to anxiety during the pandemic, especially in European continent (see doi: 10.3390/ijerph191911929). These issues were not included in the analysis but they must be, at least, discussed.

Reviewers' comments:

Reviewer's Responses to Questions

**Comments to the Author**

1. Is the manuscript technically sound, and do the data support the conclusions?

Reviewer #1: Yes

Reviewer #2: Yes

2. Has the statistical analysis been performed appropriately and rigorously? 

Reviewer #1: Yes

Reviewer #2: I Don't Know

3. Have the authors made all data underlying the findings in their manuscript fully available?

Reviewer #1: No

Reviewer #2: No

4. Is the manuscript presented in an intelligible fashion and written in standard English?

Reviewer #1: Yes

Reviewer #2: Yes

5. Review Comments to the Author

Reviewer #1: Thank you for the opportunity to review “The level of health anxiety before and during the COVID-19 pandemic”. In the present study, authors investigated the levels of health anxiety before and during the COVID-19 pandemic in an adult, working population in Norway. They didn't observe any significant alteration in health anxiety scores between the pre-pandemic period and the COVID-19 pandemic period. This paper is well written. I have only minor change to further improve the quality of this paper.

1) In the introduction, I suggest expanding the topic of HA in relation to COVID-19 with studies on different populations, as this is later discussed as a limitation of the research

2) I suggest specifying the month of the period referred to in 2019 to understand how broad this period is and to subsequently evaluate the limits of the temporal ranges of the periods under examination.

3) I would suggest adding the implications of the study that are missing.

4) I suggest adding in limits paragraph that many variables that may influence health anxiety during the pandemic period were not considered, such as the risk of contracting or having contracted the virus, or having family members who were ill with Covid-19

Reviewer #2: The manuscript is very interesting and well-written, and the topic is very relevant.

However, it is my belief that the paragraph “Statistical analysis” in the section “Material and Methods” should be clearer, with a more in-depth explanation.

The authors wrote that there were few participants with two or more measurements (n=353), but it is not specified if this number is representative of the population considered in the study.

6. PLOS authors have the option to publish the peer review history of their article (what does this mean?). If published, this will include your full peer review and any attached files.

Reviewer #1: **Yes: **Benedetta Barchielli

Reviewer #2: No

---

## [Author Response · Author response to Decision Letter 0]

19 Apr 2023

Please see the document named "Response to reviewers" as an uploaded file.

---

## [Editor Report · Decision Letter 1]

2 May 2023

The level of health anxiety before and during the COVID-19 pandemic

PONE-D-23-01947R1

Dear Authors,

We’re pleased to inform you that your manuscript has been judged scientifically suitable for publication and will be formally accepted for publication once it meets all outstanding technical requirements.

Kind regards,

Christian Napoli

Academic Editor

PLOS ONE
---

## [Editor Report · Acceptance letter]

16 May 2023

PONE-D-23-01947R1 

The level of health anxiety before and during the COVID-19 pandemic 

Dear Dr. Norbye:

I'm pleased to inform you that your manuscript has been deemed suitable for publication in PLOS ONE. Congratulations! Your manuscript is now with our production department. 

Kind regards, 

on behalf of

Dr. Christian Napoli 

Academic Editor

PLOS ONE